# Reutilizing Waste Iron Tailing Powders as Filler in Mortar to Realize Cement Reduction and Strength Enhancement

**DOI:** 10.3390/ma15020541

**Published:** 2022-01-11

**Authors:** Liyun Cui, Peiyuan Chen, Liang Wang, Ying Xu, Hao Wang

**Affiliations:** School of Civil Engineering and Architecture, Anhui University of Science and Technology, Huainan 232001, China; Cuily@aust.edu.cn (L.C.); 2018028@aust.edu.cn (L.W.); yxu@aust.edu.cn (Y.X.); 2018200211@aust.edu.cn (H.W.)

**Keywords:** mortar replacement method, improving strength, filling mortar voids, pore refinement, economic profit

## Abstract

Recently, the massive accumulation of waste iron tailings powder (WITP) has resulted in significant environmental pollution. To solve this problem, this paper proposes an original mortar replacement (M) method to reuse waste solids and reduce cement consumption. In the experiment, the author employed an M method which replaces water, cement, and sand with WITP under constant water/cement and found that the strength development can be significantly improved. Specifically, a mortar with 20% WITP replacement can obtain a 30.95% improvement in strength development. To study the internal mechanism, we performed experiments such as thermogravimetric analysis (TGA), mercury intrusion porosimetry (MIP), and SEM. The results demonstrate that the nucleation effect and pozzolanic effect of WITP can help promote cement hydration, and MIP reveals that WITP can effectively optimize pore structure. In addition, 1 kg 20% WITP mortar reduced cement consumption by 20%, which saves 19.98% of the economic cost. Comprehensively, our approach achieves the effective utilization of WITP and provides a favorable reference for practical engineering.

## 1. Introduction

In China, WITP is the most common type of tailing. Its output has reached 47.5 million tons, accounting for 39.22% of total tailings [1]. Due to the large output of tailing powder and low utilization rate of resources, people mostly continue to choose piles or landfills for treatment. These practices are simple and feasible, but they waste valuable land resources (accumulated tailings ponds) and exacerbate environmental degradation around the plant site (migration and loss of metal elements from tailings). Therefore, people have been exploring more methods to reuse tailing powder resources, such as secondary beneficiation of tailing powder and purification and separation of available metal elements in tailings with new technologies. However, the metal production of this process is low and the economic cost is high, and secondary environmental problems such as the treatment and discharge of waste tailings and waste liquid after secondary beneficiation cannot be ignored [2]. Thus, the feasibility is low.

Besides, WITP, rich in calcium and silicon elements, can be used as raw materials for the production of cement [3], brick [4], or glass ceramics [5]. However, these routes need to go through calcination, which is detrimental to the carbon footprint and energy saving. It is more meaningful for practical engineering to add iron tailings powder directly into mortar or concrete. Generally, large-size iron tailings of 0.5~4.75 mm are used as concrete aggregate, Lv et al. [6] employed iron ore tailing aggregates (IOTA) to replace normal aggregate (NA) in dam concrete. And chloride ion penetration, sulfate erosion and wear resistance tests showed that the IOTA concrete possess better ITZ and durability due to high strength and angle shape of IOTA. The iron tailings with smaller size (0.003~0.5 mm) are more inclined to replace fine aggregate. Zhao et al. [7] demonstrated that when replacement is less than 40%, for 90 days standard cured specimens, the mechanical properties of the tailings mixes is equivalent to that of the control group. Tao and Dang [8] believed that the content of iron tailings powder is low (less than 50%), and there is a positive correlation between concrete strength and content. If it exceeds 50%, the mechanical properties of concrete will decline. Iron tailings powder with a particle size less than 0.003 mm is often used as cementitious material. Ince [9] used abandoned gold tailings (GT) to improve the compressive strength and durability of mortar, including water permeability, frost resistance, and carbonation resistance; effectively solidified heavy metals of a GT. 30% substitution rate can reduce 22% of CO_2_. Obviously, the direct application of tailings to concrete or mortar can improve performance and reduce carbon. However, this does not apply to most tailings.

Bezerra et al. [10] researched that 5% replacement of iron-rich ore tailing (IOT) maintained mechanical properties within 90 days of age, but 15% IOT paste strength was reduced by 13.14%. Although the early stage promoted the hydration of the aluminate phase, the cement content decreased to 85%, which resulted in the decrease of the hydration gel product caused by the dilution effect of cement, resulting in the decrease of the mechanical properties of IOT paste. Tao and Dang [8] explained that cement is replaced by iron tailing powder, and the concrete strength tends to decline. This is caused by the reduction of CSH formation in the early stage and the alkalinity in the pore solution is not enough to activate the activity of iron tailings.

The fineness of iron tailings powder, less than 150 μm, is more inclined to choose sand replacement and cement replacement. As stated in the previous work, sand replacement is beneficial to mechanical properties and reduce natural river sand mining [11,12,13]. Cement substitution may damage mechanical properties without calcination or other activation ways [14,15]. It is unwise to improve the activity of iron tailings powder by calcination to compensate for the loss of strength caused by iron tailings replacing cement, since this behavior will erase the advantage of carbon emission reduction brought by less cement consumption. The paste replacement method may solve this contradiction. The design concept of paste replacement method is that WITP replaces cement and water at the same time, so as to ensure that the ratio of water to cement remains unchanged, as shown in Figure 1c. The previous experience declared that paste replacement method can effectively fill mortar voids, develop strength and reduce carbon footprint, such as granite dust [16], marble dust [17], pumice powder [18], and clay bricks [19]. These provides reference value for the effective use of solid waste powder.

Based on the sand replacement method and paste replacement method, we propose a mortar replacement method to replace the water, cement, and river sand in equal proportion. Figure 1 shows that M method can reduce river sand and cement consumption compared with S method and P method. Although the economic cost of M method is the lowest, it is more important to evaluate the workability, strength, and microstructure of WITP mortar based on the M method.

## 2. Raw Materials

The P.O 42.5 ordinary Portland cement used in this test is produced at the Huainan Cement Plant in Anhui Province, China. The tailings are derived from waste discharged from an iron smelter in Ma’anshan City, Anhui Province, China. The raw waste tailings are granular or massive, with a maximum size of about 1 cm (as shown in Figure 2). The primary tailings have a slight hardness. During the grinding process, they are first crushed into fine particles by an ordinary crusher, and then ground into powder with a vertical planetary ball mill with a ball-to-material ratio of 1:1 (grinding time is about 1 h). Lastly, a 0.15 mm square-hole screen is used for screening and drying in an oven, as shown in Figure 2.

### 2.1. The Basic Properties of Cement and Tailings Powder

Table 1 depicts the chemical composition of cement and tailings. As can be seen from Table 1, the cement mainly contains CaO (54.80%) and SiO_2_ (19.39%), while the main components of are CaO (43.80%), Fe_2_O_3_ (28.0%), and SiO_2_ (8.20%). Among them, the mineral components are mainly calcite, calcium iron oxide, quartz, and portlandite in Figure 3.

Figure 4 shows the particle size distribution of cement and tailings powder. 40% of the tailings powder is finer than 20 μm, which belongs to the medium size tailings material [20]. The particle size parameters of cement and WITP is shown in Table 2. It can be seen that the specific gravity of cement (3.08) is smaller than that of WITP (3.32), which means that WITP is heavier than cement. The particle diameters d_10_, d_50_ and d_90_ of cement are respectively 1.41 μm, 15.5 μm and 52.8 μm respectively, while those of WITP are 1.48 μm, 35.1 μm and 132 μm respectively. WITP is slightly thicker in size than cement. However, the particle diameter d_10_ (1.41 μm) of cement and d10 (1.48 μm) of WITP reveal that 10% of WITP still has ultrafine particles, which are similar to the size of cement particles

Under the condition of pH = 12 of alkaline solution, the surface potential of WITP is −25.8 mV, which indicates that WITP is also a negative charge on the surface in mortar pore solution in Figure 5.

### 2.2. Pozzolanic Activity Test of Tailings Powder

This paper refers to ASTM C618-05 for testing the strength activity index (SAI) of WITP. Three types of specimens were prepared in this experiment, as shown in Table 3. The control group is a standard cement mortar specimen with a water-cement ratio of 0.5, Group (S) is a mortar specimen that natural sand is used to replace 20% cement (90 g sand is used instead of 90 g cement). Group (T) is the tailing mixed mortar specimens with mineral powder instead of 20% cement [21,22].

The proportion of each group is stirred in a small stirring pot for 5 min and then poured into a cube mold (5 cm × 5 cm × 5 cm). After 24 h standard curing, the mold is demolded, and then the water bath curing is continued for 27 days and the corresponding compressive strength value is tested [23]. The paper can assist in the judgment of the pozzolanic activity of the WITP by the comparison of the compressive strength of the above three sets of test pieces.

Table 3 gives the data of 28d compressive strength. The strength values of the Control group, Group (S), and Group (T) are 25.85 MPa, 19.56 MPa, and 22.32 MPa, respectively. To obtain the corresponding Strength activity index (SAI), the compressive strength values of Group (S) and Group (T) were divided by the compressive strength values of the control group specimens. At last, the SAI of Group (S) is 74%, and the SAI of Group (T) is 86% (≥75%). As can be seen, WITP has a higher SAI value compared with natural sand, which has a certain effect on improving the strength of mortar.

### 2.3. Fine Aggregate Properties

The river sand is employed as mortar fine aggregate. According to GB/T 14684-2011 Sand for Building, the fineness modulus of river sand is 2.3, which belongs to medium sand. The mixed fine aggregate is a mixture of WITP and river sand. Table 4 shows the properties of natural river sand and mixed sand, including particle size distribution, apparent density, compact packing density, and compact porosity. The replacement of river sand with WITP will lead to the decrease of the proportion of large particles (greater than 0.15 mm) in the mixed fine aggregate with the increase of replacement rate. The apparent density and compact packing density of mixed fine aggregate increase with the increase of substitution rate, and the porosity decreases. These indicate that the micro particles of WITP fill the gap of poorly graded river sand, making the aggregate system dense and optimizing the grading.

## 3. Specimen Preparation and Test Methods

The cement replacement ensures the constant water/powder ratio, but the reduction of water cement ratio (cement dilution effect) will lead to a loose internal structure of the mortar and immeasurable damage to its strength. It is not appropriate to use one-component cement replacement.

In order to reduce the amount of cement, the method of constant water/cement ratio and reducing water/powder ratio (two-component paste replacement method) may enhance the strength [17,18,24]. Besides, the traditional sand replacement has a positive effect on mortar strength within a certain substitution range. Based on the above, this article innovatively proposes a three-component replacement method which involves simultaneously replacing water, cement, and sand. Hence, this paper will evaluate the three-component mortar replacement method with the one-component sand replacement method and the two-component paste replacement method based on the workability, strength, pore structure, and hydration reaction degree.

Herein, the mixing ratios of the three types of specimens and the standard mortar specimens of the control group are shown in Table 5. It can be seen from the Table 5 that in the specimens using the sand replacement method, the WITP will replace natural sand at the proportions of 5%, 10%, 20%, and 30%, respectively, while the ratio of water and cement remains unchanged. When the paste replacement method is adopted, the WITP will replace water and cement in the proportions of 5%, 10%, 20%, and 30%, while the quality of natural sand remains unchanged. For example, for the P-10% test piece, the mass of water is changed from 364 g to 328 g (reduction of 36 g); the mass of cement is changed from 728 g to 655 g (reduction of 73 g); the total mass reduced is the mixing amount of WITP (109 g), and the quality of the sand is always unchanged (1032 g).

Finally, the calculation method of the mortar substitution method is the same (that is, water, cement and natural sand are replaced at the same time according to a certain proportion).

### 3.1. The Mini-Slump Flow Test

The mini-slump flow test is used to evaluate the workability of fresh mortar [24]. The index is the average diffusion value of the flow of fresh mortar. The diameters in two perpendicular directions were measured and the average taken as the test result.

### 3.2. Preparation of Mortar Specimen

The standard preparation method of the ITWP mortar is the premise of reliable test results of performance test. Firstly, put the cement, natural river sand, and ITWP in the oven at 105 ± 5 °C for 8 h, and then take them out and place them in the laboratory to cool to room temperature. Then, weigh the water, cement, natural river sand, and ITWP in the mix proportion. Next, pour the cement and mixed fine aggregate into a clean mixing pot in turn and mix slowly for 30 s. Then add water and mix wet for 2 ± 0.5 min. Quickly pour the evenly mixed mortar into the plastic cube test mold (5 cm × 5 cm × 5 cm) and vibrate on the shaking table for 15 ± 5 s to remove bubbles in the mortar. After the surface moisture is slightly dry, scrape the mortar or floating foam higher than the mold with a scraper. Finally, place the test mold in the standard curing room (20 ± 5 °C) for curing for 24 ± 2 h. When the temperature is low, the demoulding test pieces can be appropriately extended, but not more than two days and nights. Then, demould and number each group of test pieces, take them out after curing for 6 days and 27 days, and test the properties of corresponding hardened mortar.

### 3.3. Mercury Intrusion Porosimetry (MIP)

The Mercury intrusion porosimetry experiment will be tested by using the equipment MicroActive AutoPore V 9600 Version. The sample comes from the test piece 28 days after hydration. The quality of the sample to be tested must be controlled between 1.6 and 2.1 g to prevent the excessive mass of the test piece from affecting the porosity test. The pressure value of the equipment needs to be set in the range of 0.1 to 61,000 psia, and the contact angle is 130°. The characteristics of internal pores of different types of specimens after 28 days of hydration will be obtained through the MIP test.

### 3.4. Thermogravimetric Analysis (TGA)

Thermogravimetric analysis is one of the more accurate methods for calculating the calcium hydroxide content in cement paste. The value of the degree of cement hydration can be calculated based on the formula. The thermogravimetric analysis equipment is TGA Q500. During the experiment, the sample temperature is controlled from 20 °C to 860 °C, the heating rate is 20 °C/min, and the nitrogen environment is 50.0 mL/min. TGA usually calculates the percentage of loss so that samples of different quality can be compared and analyzed. This paper can get the degree of hydration reaction of different types of specimens at different ages based on the TGA test.

## 4. The Results of the Experiment and Discussion

### 4.1. Workability

In the sand replacement method, the workability of fresh mortar decreases slightly with the increase of the WITP substitution rate. However, the paste replacement method and mortar replacement method show that the increase of WITP leads to the significant decrease of the workability of fresh mortar. It should be noted that when the mortar replacement rate is 30%, the blending of WITP leads to too low consistency of the mixture, as shown in Table 6. So, it is difficult to prepare uniformly test piece. Therefore, the preparation of M-30% specimens and subsequent experiments were not carried out.

### 4.2. Compressive Strength

The test specimens of the control group and the specimens prepared by the 3 replacement methods of WITP were tested for compressive strength after 28 days of curing. The final test results are shown in Figure 6. The compressive strength of C is 43.3 MPa. The replacement rate increases, and the compressive strength of different replacement methods (sand replacement, paste replacement, and mortar replacement) in the experimental group also show an upward trend. It shows that the incorporation of WITP is beneficial to the improvement of the compressive strength of the test piece.

In the condition of the same replacement rate, the specimens prepared by the mortar replacement method obviously have higher compressive strength than the paste replacement method or sand replacement method. In Figure 6, when the replacement rate is 5%, the compressive strength of the M-5% specimen prepared by the mortar replacement method is increased by 10.7% and 6.9% compared with P-5% and S-5%. Compared with P-10% and S-10%, the compressive strength of M-10% is increased by 22.3% and 15.8% respectively; when the replacement rate is 20%, the compressive strength of M-20% is compared with that of P-20% and S-20% increased by 19.1% and 14.8%, respectively. When the replacement rate exceeds 20%, the specimen prepared by the mortar replacement method cannot be prepared due to the mixing of more tailings powder. However, P-30% and S-30% can be prepared, and the compressive strength of the prepared specimen is further improved (Figure 6).

We discuss the mechanism of mortar replacement method (replacing water, cement and river sand at the same time) to promote strength development:Optimization of aggregate interface: optimizing the grading of river sand, increasing the stacking compactness of fine aggregate (as shown in Table 4), filling the interface between river sand and matrix, and optimizing the pore structure [10] (as shown in Section 4.3), are the improvement mechanism of mortar substitution.Mitigation of cement dilution effect: the reduction of water/binder ratio is conducive to improving the compressive strength of mortar (Figure 7). The constant water/cement ratio will not cause the obvious cement dilution problem of single cement substitution. The damage of cthe ement dilution effect to strength is immeasurable, unless the substitute has high pozzolanic activity [14,15].Nucleation effect: WITP provides more nucleation sites for calcium ions in pore solution. In addition, in an alkaline environment, the surface charge of WITP is significantly negative, providing electrostatic force for more calcium ion adhesion, making CSH more dense and increasing matrix strength [24].Prospective filling effect: it shows reducing the quality of paste and increasing the packing density of aggregate. Although the reduction of cement consumption will lead to the corresponding reduction of hydration products (as proved in Section 4.4), the smaller space to be filled by slurry with the increase of WITP [25,26], and WITP paste still effectively fill the internal gap of mortar [27]. Microscopically, it can improve the internal void structure and pore structure of mortar at different ages (as can be proved in Section 4.3).Activity effect of WITP: in the later stage, the pozzolanic activity of iron tailings powder also promotes the further improvement of mortar strength. (Section 2.2 prove that iron tailings powder is active).

### 4.3. Pore Structure

#### 4.3.1. Pore Diameter Distribution and Pore Characteristic

We divided the pores in the test piece according to the pore size. The pore size of 5 nm~10 nm belongs to the gel pore, which is the pores filled by the dense and disordered CSH gel generated by the internal hydration product. The pore size of 10 nm~100 nm belongs to the transition pores, which are the pores of the external hydration product CSH gel or the looser pores filled by other hydration products such as calcium hydroxide and ettringite. The pore size of 100 nm~1 μm belongs to the capillary pores. With the continuous progress of hydration, the free water space is reduced, as well as the pores that are not filled by hydration products and other solid particles. The capillary pores are usually closely related to the mechanical properties and durability [28]. The pore size of the macropores are greater than 1000 nm, and the macropores have a pore structure with obvious defects inside the mortar, which significantly affects the mechanical properties of the mortar [29].

When the replacement rate is 20%, the compressive strength of the specimens prepared by the three types of replacement methods is relatively high. Therefore, in the MIP test, the internal pore structure of the three types of specimens and the control specimens are mainly compared and tested. The result is shown in Figure 8. Compared to the control group C, S-20%, and P-20%, the interior of the test piece M-20% is denser and the overall pore structure is better. The average pore size A_v_, the most probable pore size M_p_ and the porosity of M-20% are 21.93 nm, 40.2 nm, and 7.34%, respectively, which are the lowest values. In addition, various pore types of M-20% are reduced, especially the pores between transition pores and capillary pores are greatly reduced, which makes make the most probable pore size further reduced. The optimization of the pore structure of M-20% be conducive to enhancement of mechanical properties.

Figure 9 indicates the internal pore size distribution characteristics of the three types of specimens (S-20%, P-20%, and M-20%) and the control specimen (C). It can be seen more intuitively from the figure that the internal pore structure of the M-20% specimen is the densest (in particular, the gel pores, transition pores, and capillary pores are significantly less than the control group C, S-20%, and P-20%). The macro pores are slightly less than C, S-20%, and P-20%, and there is less difference. It can be shown that when the replacement rate is 20%, the inside of the specimen prepared by the mortar replacement method has fewer pores of different dimensions (especially small-sized holes), so the corresponding mechanical properties are also better (with the above the analysis results are consistent). Besides, the control specimen C and the three types of specimens S-20%, P-20%, and M-20% using different replacement methods have corresponding total pore volumes of 0.0958 mL/g, 0.0875 mL/g, 0.0829 mL/g, and 0.0355 mL/g, respectively, in Figure 10.

According to the analysis of strength results in Section 4.2, compared with C, S-20% mainly reduces the macropore and macropore volume (greater than 0.1 μm), the transition hole is slightly lowered. This is due to the micro size effect of iron tailings powder (less than 0.15 mm), optimizing the mortar aggregate gradation and reducing the porosity. Increased aggregate bulk density (as demonstrated in Section 2). Additionally, WITP belongs to porous structure material and has water absorption characteristics. As a fine aggregate, it is mixed into the mortar to reduce the fluidity of the fresh mortar (as proved in Section 4.1). In the early stage of hydration, it can absorb the excess free water around the WITP, reduce the local water cement ratio at the interface of iron tailings powder, and enhance the interface cementation [30]. Although the total pore volume of P-20% is similar to S-20%, the capillary pore and macropore volumes are greater than S-20%. The compressive strength of P-20% is greater than S-20%. M-20% combines the advantages of P-20% and S-20% to optimize the internal pore structure of mortar most effectively. The total pore volume of M-20% is the smallest, which is 62.94%, 59.43%, and 57.18% lower than that of specimens C, S-20%, and P-20%, respectively. In addition, this paper attempts to introduce pore fractal dimension to further analyze the characteristics of pore structure.

#### 4.3.2. Fractal Dimension Analysis

In addition to the pore size distribution and pore characteristics that can describe the pore structure, the fractal dimension is also one of the pore structure parameters, which mainly characterizes the roughness and complexity of the pore structure, and this parameter is closely related to the strength [31]. It is reasonable to use Zhang’s model to estimate the fractal dimension of porous media [32]. Zhang’s [33] model suggests that the work carried out by the external force on the mercury is equal to the surface energy of mercury entering the pores, as shown in Equation (1). Where *P* represents the pressure applied to the mercury (psia), *V* means the pore volume (mL), *σ* is the mercury surface tension (N/m), *θ* represents the contact angle between the mercury and the sample, and *S* expresses the surface area of the hole to be tested (m^2^).
(1)$$\int_{0}^{v}P\cdot V=-\int_{0}^{s}\sigma \cos\theta dS$$
(2)$$\overset{n}{\underset{i=1}{\sum }}p_{i}\Delta V_{i}=C\cdot r^{2-D}\cdot V_{n}^{D/3}$$
(3)$$\log(\frac{W_{n}}{r_{n}^{2}})=D\times \log Q_{n}+C$$
(4)$$W_{n}=\overset{n}{\underset{i=1}{\sum }}p_{i}\Delta V_{i},\;\;\;\;Q_{n}=\frac{V_{n}^{1/3}}{r_{n}}$$

The surface area *S* in the Equation (1) is replaced by the pore size *r* and the pore volume *V* into a discrete formula, and thereby we obtain Equation (2). The logarithm on both sides is taken to obtain Equation (3), where *W_n_* is the intrusion work. It can be seen from Equation (3) that the fractal dimension *D* is the slope value of which can characterize the roughness and complexity of irregular pores. Generally, the fractal dimension *D* of porous media materials with fractal characteristics is between 2 and 3. When *D* is 2, it indicates that the pore surface is smooth [34]. If the value of *D* increases, and the porosity and average pore size of the porous material are found to be correspondingly reduced, this indicates that the pore structure of the porous material is more complex and irregular, and the space filling rate is increased.

Zhang [35] reveals that the scale dependence of fractal dimension is universal (that is, the fractal dimension values of pores of different sizes are different). Figure 11a–d shows the fractal dimensions of the internal pores of C, S-20%, P-20%, and M-20% after 28 days of hydration. It can be seen from the figure that the fractal dimensions of specimens C, S-20%, P-20%, and M-20% are between 2.5 and 3.0, and the linear regression correlation coefficient should be controlled at R^2^ ≥ 0.99, which can guarantee the fractal dimension validity of dimensionality. In addition, the fractal dimension of the control group C is size-dependent, and the fractal dimensions corresponding to the pore size > 5 μm and <3 μm are 2.524 and 2.791, respectively, in Figure 11a. The roughness and complexity of macropores (>5 μm) are smaller than that of small pores (<3 μm). There is an obvious jumping interval between the pore size of 3 μm~5 μm (indicated by the red circle). The main reason may be the ink bottle-shaped pores and obviously defective pores in the specimen. This interval is also the basis for dividing the pore size interval.

The red circles in Figure 11b–d have a tendency to converge toward the regression line, indicating that there is no obvious jump interval in the specimens S-20%, P-20%, and M-20%. The main reason is that the filler technology function of the WITP fills most of the ink bottle-shaped pores and defective pores, thereby optimizing the pore structure. This is consistent with the previous pore structure analysis conclusion. Moreover, the similar reports could be found in [34].

Additionally, the fractal dimensions of the entire pore interval corresponding to S-20%, P-20% and M-20% are 2.996, 2.985, and 3.001, respectively. Compared with the control group C (D = 2.524 (>5 μm) and D = 2.791 (<3 μm)), S-20%, P-20%, and M-20% simultaneously increase the roughness of the macropore pore interval (>5 μm) and small pores interval (<3 μm).

The increase in the roughness and complexity of the macroporous pores is mainly due to the filler technology of the tailings powder. Since the content of sub-micron WITP particles is small, the filling effect of small pores is low. As a result, the fractal dimensions of S-20%, P-20%, and M-20% are only slightly higher than those of the control group for small pores interval (<3 μm).

It is clearly that the fractal dimension of M-20% is the largest (3.001), indicating that the internal pore structure is the roughest in M-20%, and the filling effect of WITP is the most obvious. Jin [36] suggested that the compressive strength of mortar specimens increased with the increase of fractal dimension, and the data in this paper reached the same conclusion. In summary, according to the fractal dimension value, the following conclusions can be drawn. (1) The incorporation of WITP will reduce ink bottle-shaped pores or obvious defective pores. Under the same replacement rate, the mortar replacement method can eliminate more ink bottle-like pores or defective pore, and make the pore structure more stable. (2) It can be seen from Table 7 that the fractal dimension and strength are positively correlated. The fractal dimensions of specimens C, P-20%, S-20%, and M-20% are gradually increasing, and the corresponding compressive strength is also gradually increasing.

Figure 12 is a scanning electron microscope image of specimens C, S-20%, P-20%, and M-20%. It can be seen from the figure that there are large cracks, large pores (>1 μm) as well as capillary pores (<1 μm) composed of needle-like ettringite in the control group C. There are no obvious cracks in the sample S-20%, only a few large pores (about 1 μm) and capillary pores composed of tightly arranged ettringite. There are only tiny capillary pores (<1 μm) in sample M-20%, which are composed of denser fibrous CSH or dense clustered CSH gel accumulation, and the pore structure is the best. The above-mentioned electron microscope images show that the pore structure is consistent with the MIP test conclusions, which can fully explain that the incorporation of WITP can optimize the pore structure.

### 4.4. TGA

In the above experiment, the cement content of the control group C and S-20% did not decrease, while the P-20% and M-20% were both replaced with WITP by 20% (that is, the cement consumption was reduced by 20%). To explain the effect of cement reduction on the internal hydration reaction and product formation of the specimen, this paper will test the difference between the two sets of cement paste specimens in the hydration process based on TGA. Among them, the control group C is a cement paste specimen with a water-cement ratio of 0.5, which is represented by the symbol C-0.5; after paste replacement with 20% tailings powder, the experimental group is a specimen. It is represented by the symbol P-0.5.

TGA was adopted to calculate the amount of evaporated water and calcium hydroxide, which could evaluate to the hydration characteristic of C-0.5 and P-0.5. Figure 13 and Figure 14 are the TG-DTG curves of C-0.5 and P-0.5 at 7d and 28d, respectively. It can be seen from Figure 13 that at the seventh day of hydration, the positions of the absorption peaks in C-0.5 and P-0.5 are basically the same, and only the peaks in the range of 600~800 °C are misaligned. In Figure 14, the peak shapes of the C-0.5 and P-0.5 are consistent and there is no peak shift phenomenon at 28d of hydration. Among these absorption peaks, the temperature range of 30~200 °C is the main peak range of the first stage, and the temperature range of 30~105 °C is mainly the overflow of evaporated water and part of bound water. CSH gel decomposes and loses bound water (100 °C). 110~170 °C is mainly due to the decomposition of gypsum and ettringite and partial water loss of aluminate, while all evaporated water is removed at 120 °C [37]. Besides, there is no new peak shape in the temperature range of 30~600 °C (Figure 13 and Figure 14), indicating that there is no amorphous C-S-H gel in P-0.5 that is different from C-0.5. The temperature range of 400~550 °C is the main peak range of the second stage, mainly dehydroxylation of the calcium hydroxyde. Calcium hydroxide (CH) is the only independent peak, which is clearly different from other hydration products such as ettringite (AFt phase), AFm, and C-S-H gel.

TGA is relatively accurate for calculating the amount of CH, so this article calculates the content of evaporated water the content of calcium carbonate corrected calcium hydroxide content and the degree of hydration per cement particle separately. The specific calculation formulas are as follows:(5)$$M_{\mathrm{cc}}\left(\%\right)=\frac{\frac{\left(\Delta_{600{}^{\circ}C}-\Delta_{800{}^{\circ}C}\right)}{M_{0}}}{\frac{44.01}{100.09}}$$
(6)$$M_{\mathrm{ch}}\left(\%\right)=\frac{\frac{\left(\Delta_{400{}^{\circ}C}-\Delta_{550{}^{\circ}C}\right)}{M_{0}}}{\frac{44.01}{100.09}}+\frac{\frac{\left(\Delta_{600{}^{\circ}C}-\Delta_{800{}^{\circ}C}\right)}{M_{0}}}{\frac{44.01}{74.09}}$$
(7)$$\alpha =\frac{M_{\mathrm{ch}}}{M_{\mathrm{cement}}}\times 100\%,\;M_{\mathrm{cement}}=\frac{m_{\mathrm{cement}}}{M_{0}}$$
here, Equation (5) is the formula for calculating the content of calcium carbonate (*M*_cc_). Equation (6) is the formula for calculating the content of calcium hydroxide. 1 mole of calcium carbonate is carbonized to produce 1 mole of calcium hydroxide; 1 mole of calcium carbonate decomposes 1 mole of calcium oxide and 1 mole of carbon dioxide.

Table 8 calculation results of CH and unit hydration degree of samples C-0.5 and P-0.5 with curing for 7 days and 28 days. The pozzolanic effect of WITP is not obvious at the curing age of seven days. Here, it is considered that the CH generated at this time comes from cement hydration and is not consumed by WITP. CH contents of C-0.5 (7d) and P-0.5 (7d) are approximately equal to 21.95% and 21.07%, respectively, and the hydration degree of unit cement is 32.93% and 39.49%, respectively. The cement dosage of P-20% is reduced by 20%, but the hydration degree of the P-20% unit cement is higher. This is due to the fact that the addition of WITP can disperse the cement, provide more crystal nucleus sites for cement hydration, and promote the early hydration of cement. The hydration degree of unit cement of C-0.5 (28d) and P-0.5 (28d) are 35.92% and 37.42%, respectively. Meanwhile, the CH content of P-0.5 (28d) was lower than that of P-0.5 (7d). In the later stage, part of calcium hydroxide is consumed by iron tailings powder in P-0.5 and there is pozzolanic effect. Even so, the hydration degree of unit cement of P-0.5 (28d) is still higher than that of C-0.5 (28d).

Figure 15 shows that more needle-like C-S-H or rod-shaped ettringite is generated inside the C-0.5 specimen and that the structure is relatively loose, while the inside of P-0.5 is more densely packed C-S-H. The role of WITP in slurry replacement is mainly physical nucleation in the early stage and volcanic ash reaction in the later stage. Of course, the constant water cement ratio ensures that the strength is not damaged, and the filling effect of iron tailings powder is also one of the reasons. These can also be explained by the fact that, although the amount of cement in the sample M-20% was reduced in the previous experiment, the internal pore structure of the M-20% was the best, and the obtained compressive strength value was also the highest.

### 4.5. Cost-Friendliness

At last, this paper evaluates the economic effects of different substitution methods. The price cost of cement and river sand are provided by Ince [9], which is used to calculate 1 kg of fresh mortar for different groups, as shown in Figure 16a. When the market price of river sand is low, under the same substitution rate, the paste replacement method and mortar substitution method have approximately the same economic cost-effectiveness, and are significantly lower than sand substitution and control group. The cost price of 1 kg mortar with m-20% can be reduced by 19.96% compared with the control group.

Figure 16b is based on the survey data provided by Kim [38] to calculate 1 kg fresh mortar of different groups, respectively. When the cost of mixing water and river sand is larger, it is easier to highlight the cost advantage of mortar replacement method. The cost price of S-20%, P-20%, and M-20% 1 kg mortar decreased by 11.48%, 8.47%, and 19.98%, respectively, compared with the control group.

## 5. Conclusions

In this paper, three proportioning calculation methods, namely the sand replacement method, paste replacement method, and mortar replacement method, are used to mix the ground WITP into cement-based materials for composite mortar test-piece. The influence of WITP on the concrete and the difference of the three alternative methods are analyzed from the aspects of consistency, compressive strength, internal pore structure characteristics, and hydration reaction degree. The main conclusions obtained are as follows:Compared with the paste replacement (P) method and sand replacement (S) method, the mortar replacement (M) method is more conducive to the enhancement of early strength.The compressive strength of M-20% WITP mortar increased by 30.95%, 14.78%, and 19.12%, respectively, compared with C (control group), S-20% and P-20%.M-20% WITP mortar realized an economy of 20% cement and 20% river sand, which effectively decreased the consumption of high-energy consumption production and non-renewable resources.As expected, the P method can effectively alleviate the barrier of traditional cement replacement (cement dilution) due to the constant water/cement ratio. The addition of iron tailings powder reduces the water/binder ratio and fills voids of aggregate interface.The nucleation effect can effectively promote the hydration degree of unit cement in seven days. The pozzolanic effect consumes calcium hydroxide and generates a more dense and complex pore structure. These effects jointly improve the compactness and strength of the mortar matrix.The S method optimizes the gradation of fine aggregate, increases the bulk density, and effectively reduces the porosity in sand aggregate, which is also one of the mechanisms of M method.The market price of WITP is 12 RMB/ton with excluding transportation expenses. The cost price is negligible compared with cement. The economic cost of 1 kg M-20% WITP mortar can be reduced by nearly 20%.

## Figures and Tables

**Figure 1 materials-15-00541-f001:**
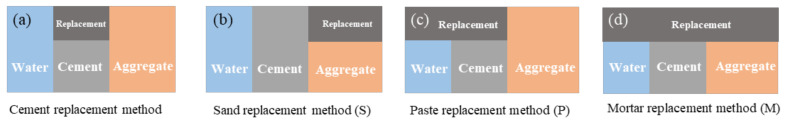
The four types (**a**–**d**) of replacement methods of tailings powder.

**Figure 2 materials-15-00541-f002:**
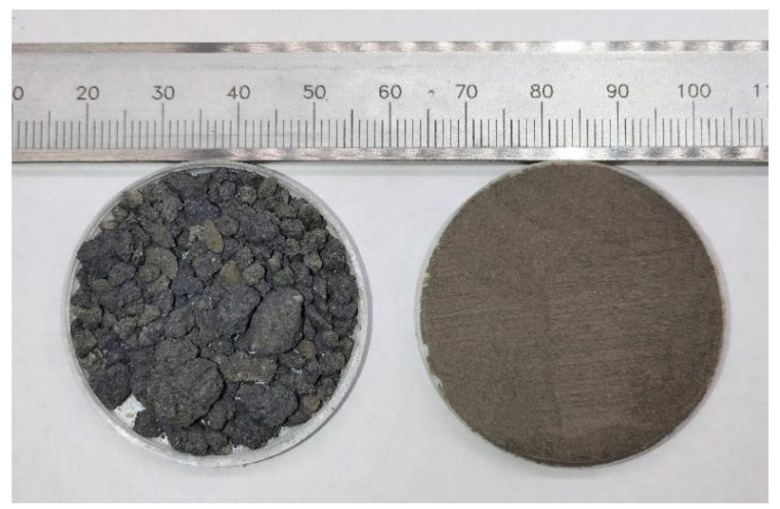
The WITP and primary tailings samples.

**Figure 3 materials-15-00541-f003:**
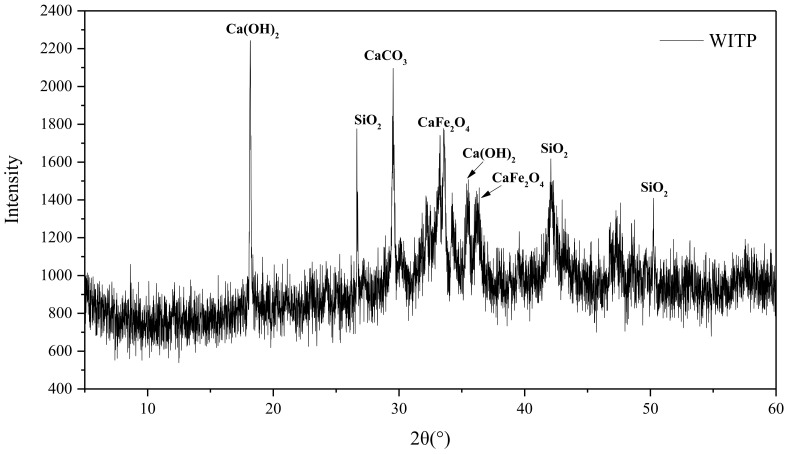
X-ray diffractograms of tailings powder.

**Figure 4 materials-15-00541-f004:**
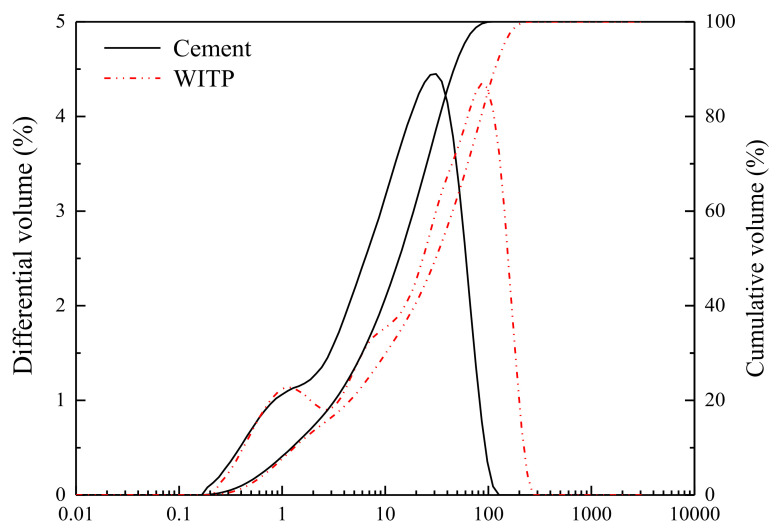
Particle size analysis of cement and tailing powder.

**Figure 5 materials-15-00541-f005:**
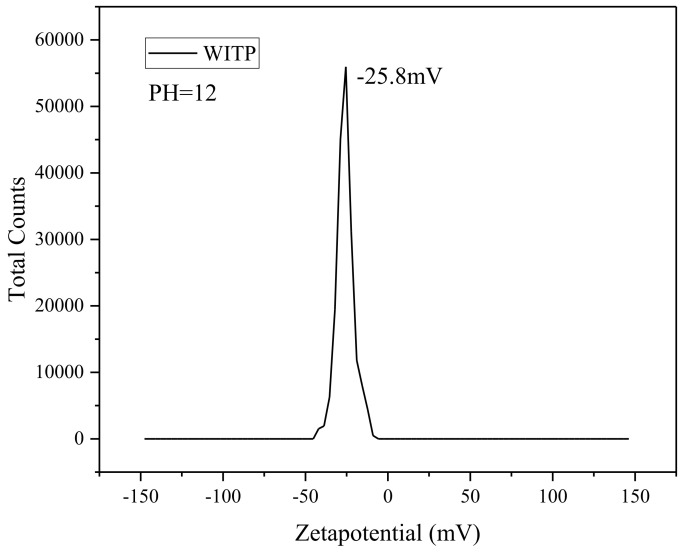
Electrical susceptibility of WITP in pH = 12 solution.

**Figure 6 materials-15-00541-f006:**
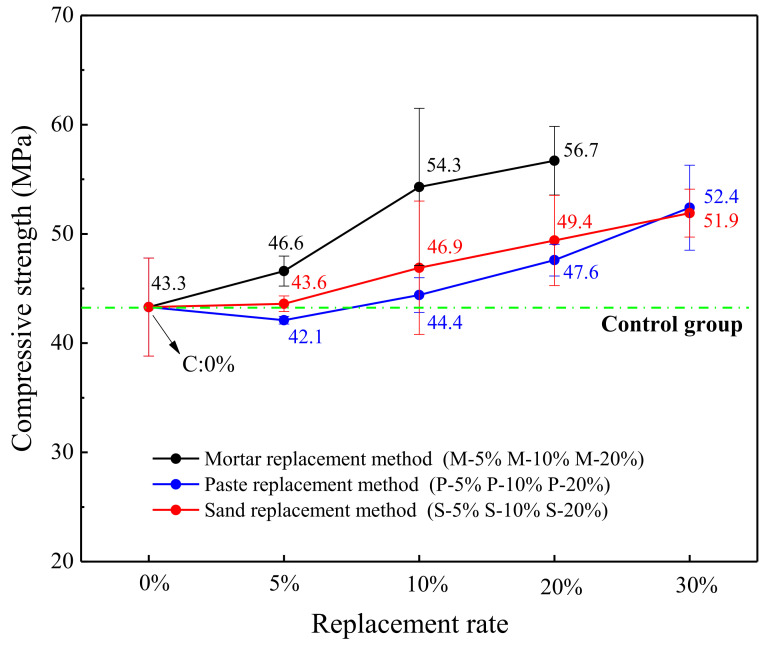
28d compressive strength of specimens prepared by different alternative methods of tailings powder.

**Figure 7 materials-15-00541-f007:**
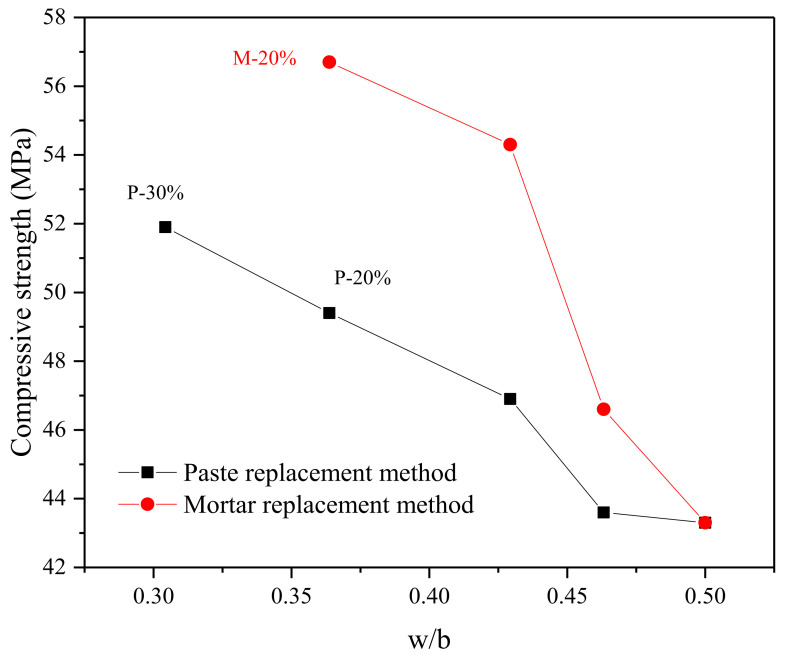
The compressive strength of WITP mortar corresponding to different water/binder ratio.

**Figure 8 materials-15-00541-f008:**
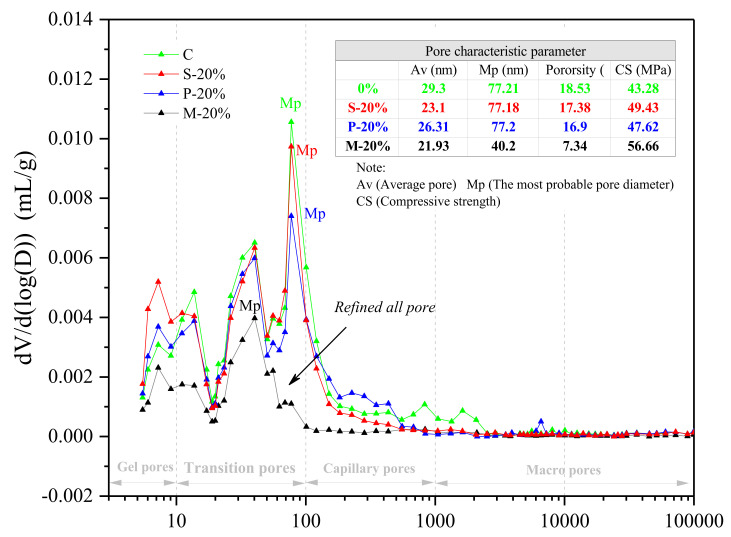
MIP test results of C S-20%, P-20%, and M-20%.

**Figure 9 materials-15-00541-f009:**
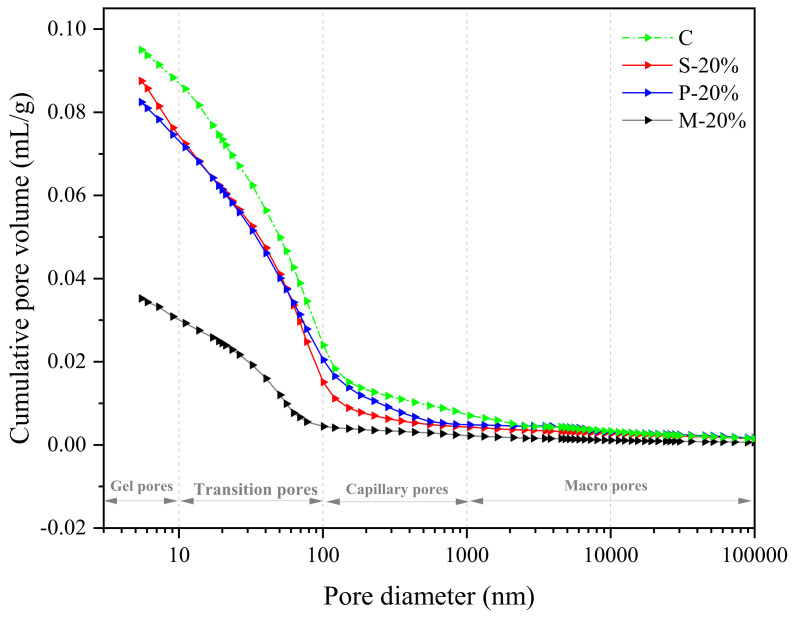
The distribution characteristics of internal pore size of C S-20%, P-20%, and M-20%.

**Figure 10 materials-15-00541-f010:**
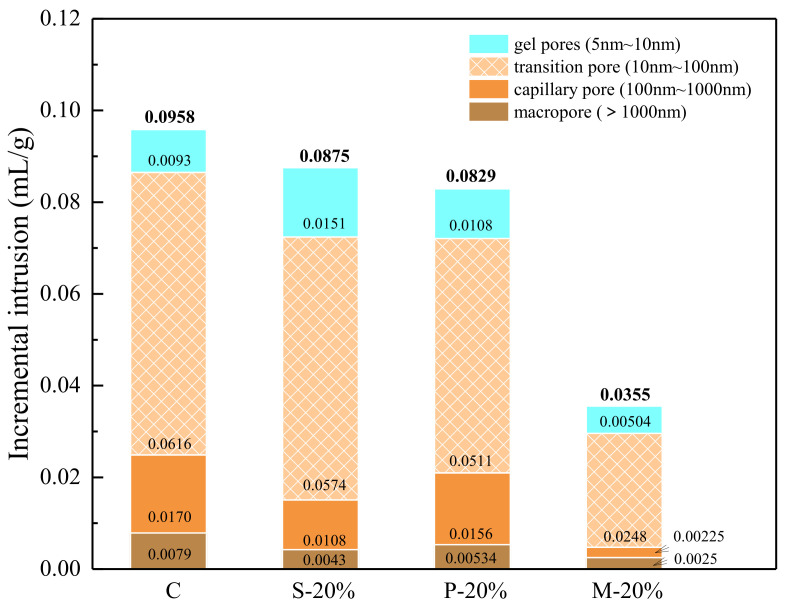
Pore volume of C S-20%, P-20%, and M-20%.

**Figure 11 materials-15-00541-f011:**
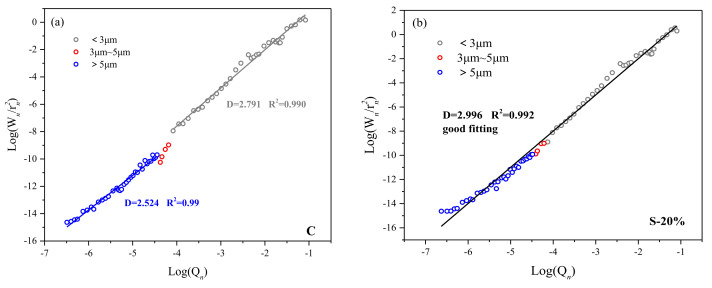
(**a**). The fractal dimension of the pores of C; (**b**) the fractal dimension of the pores of S-20%; (**c**) the fractal dimension of the pores of P-20%; (**d**) the fractal dimension of the pores of M-20%.

**Figure 12 materials-15-00541-f012:**
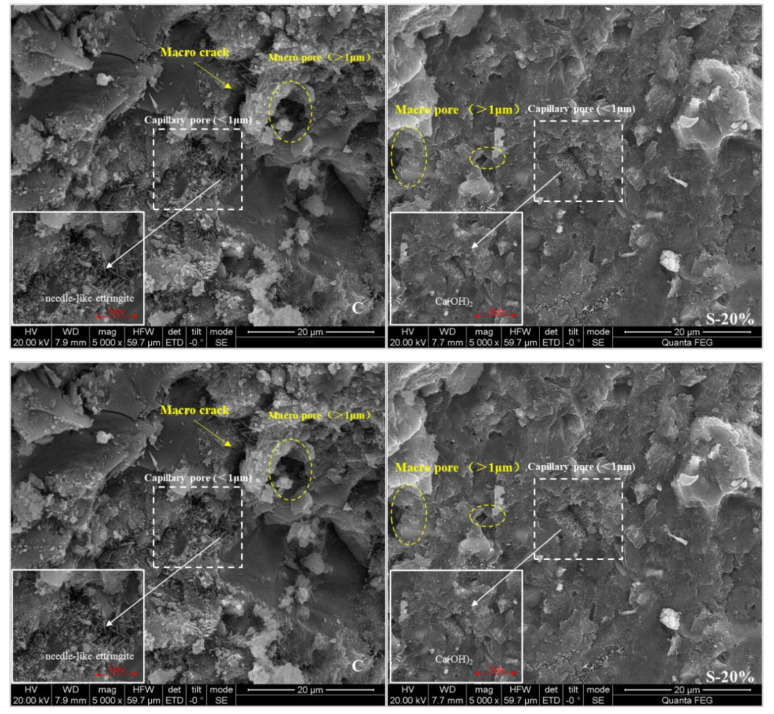
SEM images of the third type of specimens with a replacement rate of 20% and the internal specimens of the control group.

**Figure 13 materials-15-00541-f013:**
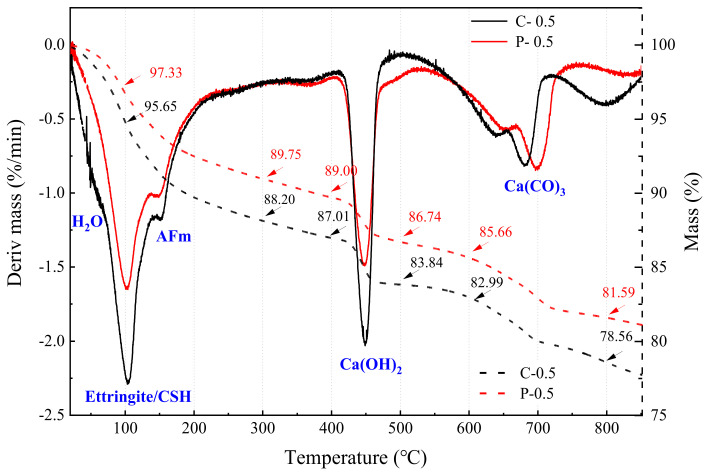
TG-DTG curves of C-0.5 and P-0.5 specimens (curing for seven days).

**Figure 14 materials-15-00541-f014:**
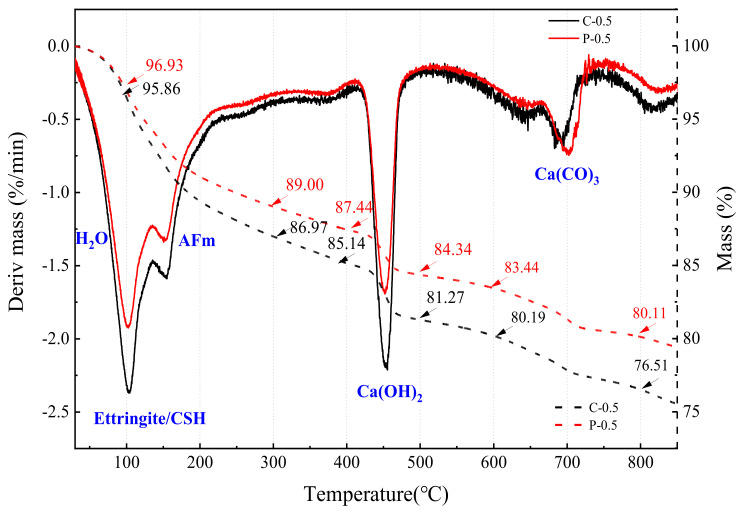
TG-DTG curves of C-0.5 and P-0.5 specimens (curing for 28 days).

**Figure 15 materials-15-00541-f015:**
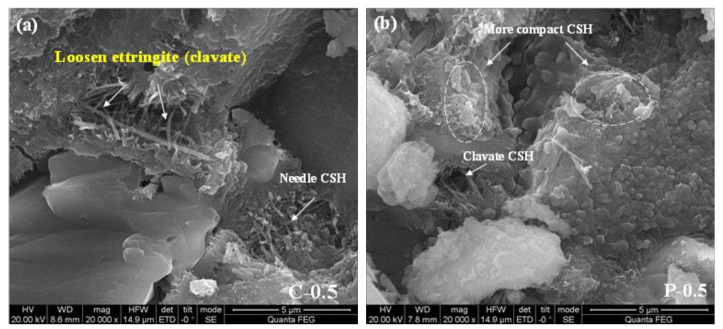
(**a**) SEM images of C-0.5 specimens after hydration for 28d. (**b**) SEM images of P-0.5 specimens after hydration for 28d.

**Figure 16 materials-15-00541-f016:**
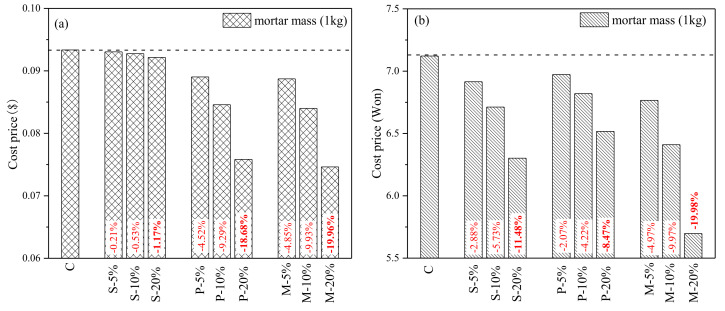
(**a**) Cost price of 1 kg mortar based on data provided by Ince [9]. (**b**) Cost price of 1 kg mortar based on data provided by Kim [38].

**Table 1 materials-15-00541-t001:** The chemical composition of cement and tailings powders material.

	Cement (wt. %)	WITP (wt. %)
SiO_2_	19.39	8.2
MgO	1.39	5.5
CaO	54.80	43.8
Al_2_O_3_	8.07	1.7
Fe_2_O_3_	2.84	28.0
K_2_O	1.00	0.1
Na_2_O	-	3.8
SO_3_	2.86	1.1
Others	9.65	7.8

**Table 2 materials-15-00541-t002:** Particle size parameters of cement and tailing powder.

Component	Cement	Tailings Powder
Specific gravity	3.08	3.32
Particle diameter d_10_ (μm)	1.41	1.48
Particle diameter d_50_ (μm)	15.5	35.1
Particle diameter d_90_ (μm)	52.8	132

**Table 3 materials-15-00541-t003:** Testing pozzolanic activity test of tailings mortar.

	Water (g)	Cement (g)	Sand (g)	WITP (g)	Compressive Strength (MPa)
Control	225	450	1350	0	25.85
Group (S)	225	360	1440	0	19.56
Group (T)	225	360	1350	90	22.32

**Table 4 materials-15-00541-t004:** Properties of natural river sand and mixed fine aggregate.

Particle Size Distribution					
Standard Square Screen (mm)	S (%)	S + 5%T (%)	S + 10%T (%)	S + 20%T (%)	S + 30%T (%)
4.75	0.00	0.00	0.00	0.00	
2.35	0.04	0.04	0.04	0.00	
1.18	2.70	4.02	3.55	4.11	
0.6	31.65	36.85	33.10	32.33	
0.3	94.36	91.23	86.67	76.35	
0.15	99.41	94.88	90.19	80.50	99.41%
**Apparent Density (kg/m^3^)**					
	S	S + 5%T	S + 10%T	S + 20%T	S + 30%T
	2618	2623	2635	2655	2698
**Compact Packing Density (kg/m^3^)**					
	S	S + 5%T	S + 10%T	S + 20%T	S + 30%T
	1750	1800	1845	1965	2067
**Compact porosity (%)**					
	S	S + 5%T	S + 10%T	S + 20%T	S + 30%T
	33.16	31.38	29.98	25.99	23.39

Note: S mean river sand, T mean iron tailings powder.

**Table 5 materials-15-00541-t005:** Preparation ratio of test pieces using different alternative methods of tailings powder.

Specimen Type	Number	Water (kg/m^3^)	Cement (kg/m^3^)	Water/Cement Ratio	WITP (kg/m^3^)	Sand (kg/m^3^)	WITP + Sand (kg/m^3^)
Control group	C	364	728	0.5	0	1032	1032
Sand replacement method	S-5%	364	728	0.5	52	980	1032
	S-10%	364	728	0.5	103	929	1032
	S-20%	364	728	0.5	206	826	1032
	S-30%	364	728	0.5	310	722	1032
Paste replacement method	P-5%	346	692	0.5	55	1032	1087
	P-10%	328	655	0.5	109	1032	1141
	P-20%	291	582	0.5	218	1032	1250
	P-30%	255	510	0.5	328	1032	1360
Mortar replacement method	M-5%	346	692	0.5	107	980	1087
	M-10%	328	655	0.5	212	929	1141
	M-20%	291	582	0.5	424	826	1250

**Table 6 materials-15-00541-t006:** The workability of WITP fresh mortar.

Methods	Substitution Rate (%)	Slump (mm)
C	0	169
	5	167
	10	165
	20	162
	30	100
Sand replacement method	5	163
	10	138
	20	104
	30	98.7
Mortar replacement method	5	157
	10	122
	20	101

**Table 7 materials-15-00541-t007:** The fractal dimension and compressive strength of the pores.

Test Specimen	Fractal Dimension	28d Compressive Strength (MPa)
C	D = 2.524 (>5 μm), D = 2.791 (<3 μm)	43.3
P-20%	2.985	47.6
S-20%	2.996	49.4
M-20%	3.001	56.7

**Table 8 materials-15-00541-t008:** The contents of each component in C-0.5 and P-0.5 specimens at the hydration of 7 and 28 days.

Test Specimen	$M_{\mathrm{cc}}(\%)$	$M_{\mathrm{ch}}(\%)$	α
C-0.5 (7d)	12.01	21.95	32.93
P-0.5 (7d)	11.70	21.07	39.49
C-0.5 (28d)	10.81	23.94	35.92
P-0.5 (28d)	9.63	19.87	37.24

## Data Availability

Not applicable.

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
