# Peer review of "Reutilizing Waste Iron Tailing Powders as Filler in Mortar to Realize Cement Reduction and Strength Enhancement"

_materials, 2022, doi:10.3390/ma15020541_

Round 1

Reviewer 1 Report

This paper is aimed at studying the use of waste iron tailings powder (WITP) as filler in cementitious mortars. In particular, the results of an experimental campaign is presented, including results of workability, compressive strength, pore structure and hydration characteristics.

The paper presents some minor deficiencies regarding written English. The general organization of the manuscript is correct. New and valuable information is provided in the paper. Conclusions are sustained by the research.

If the authors agree to consider the comments and suggestions addressed below, made with the aim of improving the final version of the manuscript, a revised version of the paper could be considered for publication in MDPI Materials Journal.

  • Abstract: Abstract is somehow confusing. First sentence is not clear nor correct. It is recommended to reorder and improve the whole text of the abstract.
  • General: The paper should be improved regarding written English.
  • Compressive Strength: no data could be found regarding the samples geometry and type of test performed. It is clear that the paper is not focused on macro mechanical properties but in micro mechanical ones. However, some mention regarding what type of compressive test was performed should be welcome.

Author Response

  • Abstract: Abstract is somehow confusing. First sentence is not clear nor correct. It is recommended to reorder and improve the whole text of the abstract.
  • Reply: The abstract has been revised, especially the first sentence, “The waste iron tailings powder (WITP), was employed as building materials, which exhibits significant superiorities in obvious economy and environmental protection.” was revised as “Recently, the massive accumulation of waste iron tailings powder (WITP) has caused environmental pollution. To solve this problem, this paper proposes an original mortar replacement (M) method to reuse waste solids and reduce cement consumption.”. thank you.
  • General: The paper should be improved regarding written English.
  • Reply: In this article, the English writing has been improved. thank you.
  • Compressive Strength: no data could be found regarding the samples geometry and type of test performed. It is clear that the paper is not focused on macro mechanical properties but in micro mechanical ones. However, some mention regarding what type of compressive test was performed should be welcome.
  • Reply: The preparation of mortar specimen (section 3.2) was added to this paper to describe the types of hardened mortar and the preparation method in detail. Besides, the compressive strength in this paper belongs to uniaxial compressive strength. thank you.

Reviewer 2 Report

The research is sound. But your presentation of the material is poor and even confusing in some parts, perhaps due to the poor usage of language.  

Please improve the following: 

  • Please define the objective clearly somewhere in the beginning. What is at the end of section one is very vague. The first time I saw what the research is about was on page 7.
  • Define MIP
  • At the end the authors say on line 537 that this research did not receive any external funds. But then you thank a whole bunch of granting agencies in the Acknowledgement. Confusing.
  • While some parts such as the results section are ok, other parts need a lot of work to improve the language.
  • There are many mistakes in referencing is in correct and inconsistent in both within the text and in the list at the end. For example, line 60 starts with "Carolina Goulart Bezerra [10] researched...." which should be corrected as "Bezerra et al [10] researched...." Note that this reference has multiple authors after Bezerra and also presenting the entire name of the first author is incorrect. The list at the end also has last names with initials and fully spelled out names, inconsistently.
  • Please remove the last section on "outlook" as it is only a wish list and does not add any value to the paper.

Round 2

Reviewer 2 Report

Authors have addressed most of my comments from the previous round to a satisfactory level.